# Multistate Markov Model to Predict the Prognosis of High-Risk Human Papillomavirus-Related Cervical Lesions

**DOI:** 10.3390/cancers12020270

**Published:** 2020-01-22

**Authors:** Ayumi Taguchi, Konan Hara, Jun Tomio, Kei Kawana, Tomoki Tanaka, Satoshi Baba, Akira Kawata, Satoko Eguchi, Tetsushi Tsuruga, Mayuyo Mori, Katsuyuki Adachi, Takeshi Nagamatsu, Katsutoshi Oda, Toshiharu Yasugi, Yutaka Osuga, Tomoyuki Fujii

**Affiliations:** 1Department of Obstetrics and Gynecology, Graduate School of Medicine, The University of Tokyo, Tokyo 113-8655, Japan; aytaguchi-tky@umin.ac.jp (A.T.); tomotanaka-tky@umin.ac.jp (T.T.); BABAS-GYN@h.u-tokyo.ac.jp (S.B.); kawataa-gyn@h.u-tokyo.ac.jp (A.K.); satokojolly@yahoo.co.jp (S.E.); tsuruga-tky@umin.ac.jp (T.T.); mayumori-tky@umin.ac.jp (M.M.); kadachi-gyn@umin.ac.jp (K.A.); tnag-tky@umin.ac.jp (T.N.); katsutoshi-tky@umin.ac.jp (K.O.); yasugi-tky@cick.jp (T.Y.); yutakaos-tky@umin.ac.jp (Y.O.); fujiit-tky@umin.org (T.F.); 2Gynecology Division, Tokyo Metropolitan Cancer and Infectious Diseases Center, Komagome Hospital, Tokyo 113-8677, Japan; 3Graduate School of Economics, The University of Tokyo, Tokyo 113-0033, Japan; hara.konan@e.u-tokyo.ac.jp; 4Hematology Division, Tokyo Metropolitan Cancer and Infectious Diseases Center, Komagome Hospital, Tokyo 113-8677, Japan; 5Department of Public Health, Graduate School of Medicine, The University of Tokyo, Tokyo 113-0033, Japan; juntomio@m.u-tokyo.ac.jp; 6Department of Obstetrics and Gynecology, School of Medicine, Nihon University, Tokyo 173-8610, Japan

**Keywords:** cervical intraepithelial neoplasia, high-risk human papillomavirus, multistate Markov model, retrospective cohort study, survival analysis

## Abstract

Cervical intraepithelial neoplasia (CIN) has a natural history of bidirectional transition between different states. Therefore, conventional statistical models assuming a unidirectional disease progression may oversimplify CIN fate. We applied a continuous-time multistate Markov model to predict this CIN fate by addressing the probability of transitions between multiple states according to the genotypes of high-risk human papillomavirus (HPV). This retrospective cohort comprised 6022 observations in 737 patients (195 normal, 259 CIN1, and 283 CIN2 patients at the time of entry in the cohort). Patients were followed up or treated at the University of Tokyo Hospital between 2008 and 2015. Our model captured the prevalence trend satisfactory, particularly for up to two years. The estimated probabilities for 2-year transition to CIN3 or more were the highest in HPV 16-positive patients (13%, 30%, and 42% from normal, CIN1, and CIN2, respectively) compared with those in the other genotype-positive patients (3.1–9.6%, 7.6–16%, and 21–32% from normal, CIN1, and CIN2, respectively). Approximately 40% of HPV 52- or 58-related CINs remained at CIN1 and CIN2. The Markov model highlights the differences in transition and progression patterns between high-risk HPV-related CINs. HPV genotype-based management may be desirable for patients with cervical lesions.

## 1. Introduction

Cervical cancer is the second most commonly diagnosed type of cancer, and the third leading cause of cancer-related death among females in developed countries [1]. In 2012, an estimated 527,600 new cases of cervical cancer were diagnosed and 265,700 related deaths were reported worldwide [1]. In Japan, 33,114 women were newly diagnosed with cervical cancer in 2013 [2], and the mortality rate was 4.1 per 100,000 person-years [3]. Infection with human papillomaviruses (HPVs) is the main cause of cervical cancer development [4,5,6]. The types of HPV are categorized on the basis of their carcinogenesis. The International Agency for Research on Cancer divided the HPV genotypes into the following groups according to their carcinogenesis: the highly carcinogenic Group 1 (HPVs 16, 18, 31, 33, 35, 39, 45, 51, 52, 56, 58, and 59); the probably carcinogenic Group 2A (HPV 68); and the possibly carcinogenic Group 2B (HPVs 26, 30, 34, 53, 66, 67, 69, 70, 73, 82, 85, and 97) [7]. The high-risk HPV (hrHPV)-derived oncogenes E6 and E7 are necessary for malignant conversion. E6 and E7 inactivate the p53 tumor suppressor, and suppress the expression of retinoblastoma proteins, resulting in resistance to apoptosis and the promotion of cell proliferation [6]. The stabilized expression of E6 and E7 is the critical step in the progression of cervical cancer [8,9].

Worldwide, patients with a high-grade squamous intraepithelial lesion (HSIL) undergo surgical interventions (e.g., conization and loop electrosurgical excision or laser vaporization), regardless of the HPV genotype [10,11]. In Japan, surgical intervention to patients with CIN2 was not common, because approximately half of these patients regress within two years [12]. A prospective cohort study involving patients with cervical intraepithelial neoplasia (CIN) has demonstrated that, without treatment, approximately 20% of CIN2 patients with HPVs 16, 18, 31, 33, 35, 45, 52 or 58 progress to CIN3 within 5 years [12]. On the basis of the available evidence, excision strategies can be considered in patients with CIN2 as a risk-reducing treatment when the lesion is caused by these hrHPVs [13]. However, surgical excision in patients with HSIL occasionally leads to poor obstetric outcome, including preterm birth, low-birth-weight infants, and cesarean delivery, due to cervical incompetence after surgery [14,15]. Therefore, surgical treatment is unfavorable for patients who desire to be pregnant when their risk for progression to CIN3 or cancer is low.

Among the hrHPVs, HPVs 16 and 18 are the most frequently observed genotypes in patients with cervical cancer, and approximately 70% of cervical cancers are HPV 16 or 18 positive [4,16]. Several studies, including prospective cohort studies, demonstrated that the risk of HPV 16-positive patients to develop CIN2 or CIN3 lesions is higher than that reported in patients positive for other hrHPVs [12,17,18,19]. Matsumoto et al. revealed that seven hrHPV types, including HPV 16, show a high rate of progression of CIN1–2 to CIN3 compared with the other hrHPVs [12]. In addition, several reports demonstrated that patterns of persistent infection or persistent cervical lesions may differ according to the HPV types [20,21,22]. Collectively, we consider that the natural history of HPV-infected CIN lesions differs depending on the HPV genotypes; i.e., some HPV genotype-related lesions are likely to progress to cervical cancer, some are likely to regress, and others are likely to result in persistent disease.

The Cox proportional hazards model is most frequently used to predict the risk of transition from one state to another. However, the natural history of CIN shows a bidirectional transition between different states, and patients intricately move between a series of states. For example, CIN2 can progress to CIN3 or regress to CIN1 during the follow-up period. A model capable of estimating the risks of transitions from and to multiple states is more suitable for assessing the fate of CIN patients compared with the traditional Cox proportional hazards model, accounting only for the transition between two states. A solution is the use of multistate Markov models that enable investigators to estimate the transitions between multiple states [23,24,25]. Use of this approach has recently been adopted in various clinical settings [26,27,28,29,30,31,32,33,34].

In the present study, we retrospectively analyzed a cohort of patients with HPV genotype-confirmed CIN to investigate the relationship between HPV genotypes and clinicopathological features in CIN/cervical cancer. We applied a continuous-time multistate Markov model to independently estimate the prognosis of cervical lesions for designated HPV categories.

## 2. Results

### 2.1. Patients

Among the 1417 patients, 737 patients were enrolled with the following diagnoses at the time of entry: normal, 195 patients; CIN1, 259 patients; and CIN2, 283 patients. The median age was 37.6 years (interquartile range [IQR]: 31.8–44.5 years), the median number of visits was seven (IQR: 4–12), and the median duration of the follow-up was 3.03 years (IQR: 1.21–4.91 years). Table 1 displays the summary statistics for the combination of the defined HPV categories and diagnoses at the time of entry. In these patients, HPVs 16, 52, and 58 were the three most frequently observed HPV types. Because HPV 18 is the second most frequently observed HPV genotype in cervical cancer [35], in this study, we focused on HPVs 16, 52, 58, and 18.

Table 2 displays a summary of the transitions from each diagnosis of cervical epithelial lesions for the six HPV categories. Of the total 6022 transitions, 702, 270, 1008, and 852 transitions were observed for HPVs 16, 18, 52, and 58, respectively. The remaining 1501 and 2300 transitions were observed for other hrHPVs and no hrHPVs, respectively. For normal patients, 75–90% remained at the same state (e.g., 84.4% of HPV 16-positive normal patients at a visit were also diagnosed as normal at the subsequent visit). For CIN1 patients, 29–55% regressed to normal, 34–44% remained at the same state, and the rest progressed to CIN2 or CIN3. For CIN2 patients, >50% remained at the same state and the probability of progression to CIN3 was HPV type dependent: 15.6%, 10.3%, 11.1%, 7.7%, and 8.5% for HPVs 16, 18, 52, 58, and other hrHPVs, respectively. HPV 16-positive patients tended to shift to more severe states compared with those who were positive for the other HPV genotypes (e.g., 37% of those with CIN1 shifted to more severe states at the subsequent visit).

### 2.2. Prognosis of HPV-Infected Cervical Lesions Estimated Using the Markov Model

Table 3 shows the predicted probabilities of transitions from states to states within two years for the six HPV categories, as estimated by the Markov model. Regarding HPV 16-positive patients, 13%, 30%, and 42% of normal, CIN1, and CIN2 patients, respectively, progressed to CIN3/cancer. In contrast, 43% and 34% of CIN1 and CIN2 patients, respectively, regressed to the normal state. The fates of HPV 18-infected cervical lesions were similar to those of HPV 16-infected cervical lesions. However, exceptions to these were the probabilities for progression to CIN3/cancer: from the normal state, 7.6%; from CIN1, 15%; and from CIN2, 32%, respectively. The fates of HPV 52/58-positive cervical lesions differed considerably versus those of HPV 16/18-positive cervical lesions: one-third to two-fifths of HPV 52/58-positive patients were eventually diagnosed with CIN1 or CIN2 regardless of the initial state, and their probability of progression from CIN2 to CIN3/cancer was approximately 25%. Among the hrHPV-positive patients, those with other hrHPVs were most likely to regress to the normal state and least likely to progress to CIN3/cancer: probability of regression to normal state, 66% and 52% from CIN1 and CIN2, respectively, and probability of progression to CIN3/cancer, 3.4%, 8.4%, and 22% from normal, CIN1, and CIN2, respectively. The probability of regression to the normal state of no hrHPVs patients was higher than that of hrHPV-positive patients: 81% and 71% for CIN1 and CIN2, respectively. The Cox model yielded the progression probability from CIN1 to CIN2 or more severe lesions to be 50%, 9%, 53%, 50%, 25%, and 13% for HPVs 16, 18, 52, 58 other hrHPVs, and no hrHPVs, respectively (Appendix A).

Figure 1 demonstrates the observed and simulated prevalence transition of each state for the HPV categories. Overall, our model captured the trend of the prevalence, especially up to two years. However, as time progressed, the overestimation (underestimation) of the prevalence of the normal (CIN3/cancer) state expanded.

Appendix A display the summary statistics and the summary of the transitions from the dataset of the alternative model. Appendix A show the results of all sensitivity analyses using the dataset and categories of the alternative model. These data indicate that the results of the main model were fairly robust when we altered the study population and categories.

## 3. Discussion

This study is the first to estimate the parameters of the continuous-time multistate Markov model from one data set to shed light on the prognosis of cervical lesions based on the infected HPV types. The results indicated that the prognosis of hrHPV-infected cervical lesions differs according to the types of HPV and the grades of CIN.

The Markov model revealed that the rate of progression to CIN3 from any condition was highest in HPV 16-positive patients, followed by HPV 18-positive patients. These results are consistent with those reported in previous studies. In addition, our Markov model also revealed that HPV 52/58-positive patients tend to remain between CIN1 and CIN2. The Markov model clarifies the characteristics of HPV 52- and HPV 58-related lesions, as it demonstrates the probabilities of transitioning from state to state. Contrary to HPV 16-positive lesions, which highly progress to CIN3/cancer, HPV 52- and 58-related lesions were characterized by their stability between CIN1 and CIN2. In this study, we estimated the 2-year prognosis of patients with CIN using two types of models. The main model reflects transitions based upon objective results, and it is not always in line with clinical practice, because some patients with CIN2 undergo treatment without a final diagnosis of CIN3. On the other hand, the alternative model well reflects the trajectory of clinical practice. However, treatment is occasionally affected by the physicians’ decision and patients’ choice; therefore, the alternative model may lack objectivity. Admitting that our models both have advantages and disadvantages, we consider the results for the 2-year prognosis of CIN patients to be reasonably robust, as both models essentially yielded identical results.

The continuous-time multistate Markov model is more capable of reflecting the features of HPV-infected cervical lesions than the frequently used Cox proportional hazards model. Most of the previous studies have focused on the risk of progression to CIN3 or cervical cancer according to the types of HPV or grades of CIN, implicitly assuming a unidirectional disease progression. Considering the bidirectional nature of CIN, the Markov model has the potential to more accurately estimate the fates of HPV-infected cervical lesions than the standard Cox proportional hazards model. Indeed, the Markov model and the Cox model led to quite different predicted 2-year transition probabilities from CIN1 to CIN2 or more severe lesions. Notably, we could not detect higher disease progression risk in HPV 16-positive patients than in HPV 52/58-positive patients in the Cox model. The Markov model can estimate the transition between multiple states besides the probability of the customarily used endpoint, e.g., CIN3 and cervical cancer. The use of this model highlighted the difference between the natural history of cervical lesions infected with the focused four HPV types. We conclude that the model may improve our understanding of the natural history of cervical lesions and aid in the management of hrHPV-related lesions.

The Markov model was suitable for HPV-infected cervical lesions because the virus–host interaction (virus vs. host immune system) leads to bidirectional transitions between states. Activated host immune responses contribute to CIN regression, whereas immunosuppressive conditions promote CIN progression [36,37,38]. Likewise, this model can be applied to virus-induced persistent conditions, such as hepatitis B virus- or hepatitis C virus-infected liver conditions, and human immunodeficiency virus-related conditions. Indeed, previous studies applied the model to hepatitis C virus-related liver diseases with three states (i.e., none or mild fibrosis, moderate fibrosis and cirrhosis) to quantify differences in the prognosis among cohorts [34].

In this study, we could not fully estimate the transitions of HPV 18-positive patients with the present Markov model due to the limited sample size. HPV 18 is the second most frequently observed HPV genotype in patients with cervical cancer, whereas it is not frequently detected in those with precancer lesions [35]. In addition, the detection rate of HPV 18 is higher in adenocarcinoma than that in squamous cell carcinoma [39]. Further accumulation of cases may enable us to model the complicated transitions of HPV 18-positive patients.

This study has several limitations. First, we performed the HPV genotyping test once for each patient. Therefore, the change to negative for the HPV test (latent infection or clearance) was not taken into account. Previous reports demonstrate that patients with HPV clearance tend to regress compared with those with persistent HPV infection [40,41]. Patients with the transient HPV infection and persistent infection might take different courses. 

Second, in this study, we did not evaluate the effect of concurrent multiple HPV infections. In previous reports, coinfections with multiple α9 species were associated with an increased risk of CIN2 or more severe lesions compared with single infections [42,43]. Because there are many different possible combinations of concurrent infection, the numbers of patients for each possible combination was expected to be too small for an analysis of the effect of concurrent infection. Further analysis with a larger sample size is warranted to unveil the fates of cervical lesions with multiple HPV infections.

Third, the evaluation of model fitness uncovered that the simulated prevalence of the normal state (CIN3/cancer) was overstated (understated) compared with the observed prevalence. These discrepancies imply a violation of the Markov property, i.e., the distribution of the next states depends only upon the present state. Misclassification and omitted covariates, such as age, can be a potential source of the violation. The application of a hidden Markov model that can incorporate the possibility of misclassification through the information from the previous or more distant visits may be an intriguing direction for further research. 

Fourth, we have not assessed the validity of our model with the use of a test set separate from the training set, which is the dataset used for parameter estimation. Instead, we prioritized improving the parameter estimation accuracy by using all data for parameter estimation. This is currently a common practice in the literature [26,27,28,29,30,31,32,33,34]. If a sufficient sample size can be acquired, the external validity can be assessed to some degree using a test set, which will be an issue for future research.

Lastly, this study was based on data obtained from a single institute with some censoring. Especially, there is a selection bias; for example, patients enrolled in this study previously had abnormal cytology. In addition, some patients with relatively severe CIN2 were censored because of the treatment according to the physician’s decision. However, by conducting the sensitivity analysis, we confirmed that the impact on the results was limited. Nevertheless, as we have not assessed the external validity of our model in the general population, care must be taken when extrapolating the present results to the entire population, unless similar results are yielded from other institutes.

## 4. Materials and Methods

### 4.1. Study Design and Patients

This study was performed in accordance with the Declaration of Helsinki. Approval was obtained from the internal institutional review board (Approval number: 1390-1, G10082-7, and G0637-6) for this study.

This study was a retrospective cohort analysis of data extracted from the electronic health records of a teaching hospital in Japan. Since 2008, human papillomavirus (HPV) genotyping has been performed in the University of Tokyo Hospital (Tokyo, Japan) for outpatients. The patients who had abnormal cytology in the population-based screening, or whose abnormal cytology was found in the outpatient visits at the Obstetrics and Gynecology Department of the University of Tokyo Hospital or other hospitals, were enrolled. At the first visit, we confirmed the diagnosis by performing punch biopsy under colposcopic examination (based on the histological diagnosis). As follow-up screening, we routinely performed combination examinations of cytology and colposcopy because these examinations are non-invasive. We performed histological diagnosis only when the disease progressed. We reviewed the electronic health records of 1417 patients for whom genotyping was performed between October 1, 2008, and March 31, 2015, to construct a retrospective cohort. Patients were included in the study if they were (i) diagnosed with normal cervical lesion, CIN1, or CIN2 and (ii) observed for at least two visits during the study period. Patients were excluded if they had HPV 6- or HPV 11-single-positive lesions with only the diagnosis of condyloma during their follow-up period, or had only glandular lesions. Patients were followed up until they were diagnosed with cervical cancer, underwent treatment, transferred to another hospital, or until the date March 31, 2018, whichever came first. In our practical management, most patients with CIN2 were followed up without any treatment. Patients diagnosed with CIN3 underwent surgical intervention, such as conization and loop electrosurgical excision or laser vaporization.

### 4.2. Variables

The date of birth was extracted for each patient. Patient age was defined as their age at the time of entry. Follow-up interval was defined as the length of time between two consecutive visits.

Patient cytological and histological results and the date were recorded for each visit. Cytological and histological results were combined to classify the results into any of the following four diagnoses: normal, CIN1, CIN2, and CIN3/cancer. The investigators (experts in gynecologic oncology) convened and determined the criteria for pathological diagnosis as follows: (1) CIN1–2 was classified into CIN2; (2) CIN2–3 and carcinoma in situ were classified into CIN3; (3) uncertain diagnoses (e.g., atypical squamous cells of uncertain significance, atypical squamous cells that cannot exclude HSIL, and dysplasia without grading) were excluded from the study due to concerns regarding diagnostic reliability; and (4) in the presence of histological and cytological examinations, the most severe classification was adopted as the final diagnosis.

The results of the HPV genotyping in cervical samples collected using swabs were recorded. Genotyping was performed once for each patient; thus, the HPV type assigned to a patient did not change over time. It was permitted to assign multiple genotypes to a single patient. On the basis of the classification of the International Agency for Research on Cancer, we defined HPVs classified in Group 1 or Group 2A (HPVs 16, 18, 31, 33, 35, 39, 45, 51, 52, 56, 58, 59, and 68) as “high-risk HPVs (hrHPVs)”. Of these, HPVs 16, 18, 52, and 58 were separately categorized. hrHPVs other than HPVs 16, 18, 52, and 58 were classified as “other hrHPVs.” Patients who were not infected with any hrHPVs were referred to as “no hrHPVs” patients. Patients without HPV infection were also categorized in this group.

### 4.3. DNA Extraction and HPV Genotyping

Cervical samples were tested for HPV DNA using PGMY line-blot hybridization, as previously described [44]. DNA was extracted from cervical samples using the DNeasy Blood Mini Kit (Qiagen, Crawley, UK). HPV genotyping was performed using the PGMY-CHUV assay method. Briefly, standard polymerase chain reaction (PCR) was performed using the PGMY09⁄11 L1 consensus primer set and the human leukocyte antigen-DQ primer set, as previously described [44]. Reverse blotting hybridization was performed. Heat-denatured PCR amplicons were hybridized to probes specific for 32 HPV genotypes and the human leukocyte antigen-DQ references.

### 4.4. Continuous-Time Multistate Markov Model

We used the continuous-time multistate Markov model to estimate the prognosis of each patient with HPV-infected cervical lesions [23,24,25]. We defined the following four states: normal (state 1), CIN1 (state 2), CIN2 (state 3), and CIN3/cancer (state 4) (Figure 2). The arrows in Figure 2 specify possible transitions between the states defined in our model; all transitions between adjacent states, except the backward transition from CIN3/cancer to CIN2, were allowed. CIN3/cancer was the absorbing state.

Each transition parameter λ in Figure 2 indicates the transition intensity; i.e., λ_ij_ is interpreted as an “instantaneous risk” of transition from state i to j [24]. Identification of transition parameters in the continuous-time multistate Markov model requires observation of the corresponding transitions.

### 4.5. Dataset Construction

Initially, an unbalanced panel data with the unit of observation being one visit was constructed on the basis of the chart review. We truncated observations after the diagnosis of CIN3 or cancer to make the data used in the estimation compatible with the model in Figure 2. Additionally, we excluded samples that left only one observation after the truncation to estimate the model, as these samples do not contribute to the estimation. Figure 3 shows the transitions of typical samples from the baseline dataset, which illustrates that the patterns of disease progression or regression varied among patients.

### 4.6. Statistical Analysis

Summary statistics are reported for the combination of defined HPV categories (HPV 16, HPV 18, HPV 52, HPV 58, other hrHPVs, and no hrHPVs) and diagnoses at the time of entry (normal, CIN1, and CIN2). In cases of observed coinfection with different HPV genotypes, it was possible to include the same patient in the summary statistics of multiple HPV categories.

A maximum likelihood estimation was performed to estimate the parameters using the *msm* package in R [45]. Intuitively, the likelihood function consists of the sum of the probabilities of all possible paths given the observations, and each observed transition contributes to it. Parameters for the defined HPV categories were independently estimated. In cases of coinfection, the patients could contribute to the parameter estimation of several HPV categories. According to the estimated parameters, we simulated the probabilities of transitioning from state to state after two years. This 2-year transition probability was selected for two reasons. First, since the typical follow-up duration of this study was <5 years (upper quartile value of the follow-up duration was 4.9 years), a prediction beyond this period was unwarranted. Second, a 2-year prediction is considered a good benchmark for the prognosis of hrHPV-related cervical lesions. Notably, ≥50% of these lesions regress to the normal state, and approximately 10% of these lesions progress to CIN3 within two years [12]. The Cox proportional hazards model was also used to predict the 2-year transition probability from CIN1 to CIN2 or more severe lesions for each HPV category. To this end, we restricted the patients to those who were diagnosed with CIN1 at the time of entry and followed them until they were diagnosed as CIN2 or more severe lesions or at the end of their observation period.

Additionally, we simulated the change in the prevalence of each state over the five years for the HPV categories. To make it easier to compare the observed and the simulated prevalence, the initial prevalence of each state was set to be the same as the data used for parameter estimation. The fit of the model was assessed on the basis of the comparison of the observed and the simulated prevalence.

### 4.7. Sensitivity Analysis

Some patients with CIN2 underwent treatment without a final diagnosis of CIN3. Some patients with CIN3 were observed without any treatment if the lesion immediately regressed to CIN2 or a lower state. Therefore, we specified an alternative model that encompasses these situations with the following four states: normal (state 1), CIN1 (state 2), CIN2/CIN3 (state 3), and treatment (state 4) (Appendix A). The arrows in Appendix A specify possible transitions between these states; all transitions between adjacent states, except the backward transition from treatment to CIN2/CIN3, were allowed. Treatment was the absorbing state. We truncated observations after the diagnosis of treatment intervention in this alternative model, and excluded samples that left only one observation after the truncation to estimate the model. Typical sampling situations from this alternative dataset are illustrated in Appendix A.

## 5. Conclusions

In this study, we applied a continuous-time multistate Markov model to predict the prognosis of patients with HPV-infected cervical lesions. We demonstrated that the Markov model is a promising analytical method considering the bidirectional nature of these lesions. The study revealed that the natural history of hrHPV-related cervical lesions differed among the focused four HPV types: HPV 16-positive lesions were likely to be upgraded to CIN states in a step-by-step manner; HPV 52/58-positive lesions were likely to be maintained between CIN1 and CIN2; lesions positive for the other hrHPVs were most likely to regress to the normal state and least likely to progress to CIN3/cancer. On the basis of the present findings, HPV genotype-based management may be desirable for patients with cervical lesions.

## Figures and Tables

**Figure 1 cancers-12-00270-f001:**
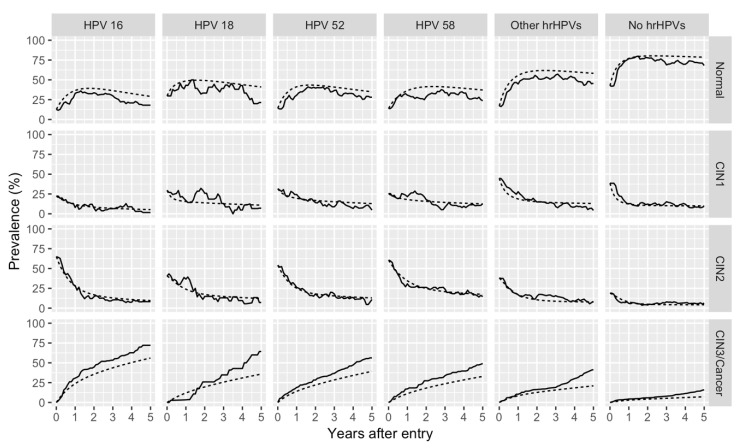
Observed and simulated prevalence transition of each state for the human papillomavirus (HPV) categories. The figure demonstrates the observed (solid line) and simulated (dotted line) prevalence transition of each state for the six HPV categories. Cytological and histological results were combined to classify the results into the states of the model. HPVs 16, 18, 31, 33, 35, 39, 45, 51, 52, 56, 58, 59, and 68 were classified as high-risk HPVs (hrHPVs). Of these, HPV 16, 18, 52, and 58 were separately categorized. hrHPVs other than HPVs 16, 18, 52, and 58 were classified as “other hrHPVs.” Patients who were not infected with any hrHPVs were referred to as “no hrHPVs” patients. In cases of observed coinfection with different HPV genotypes, it was possible to include the same patient in multiple HPV categories.

**Figure 2 cancers-12-00270-f002:**
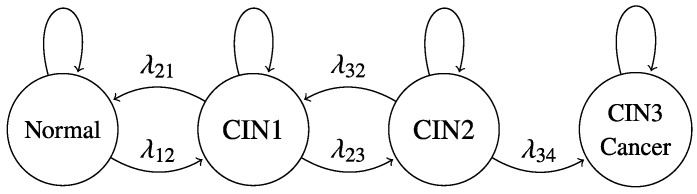
Markov model for the disease progression and regression of cervical epithelial lesions. The figure displays the schema of the Markov model for the model specified in this study. We defined four states: normal (state 1), cervical intraepithelial neoplasia 1 (CIN1, state 2), CIN2 (state 3), and CIN3/cancer (state 4). The arrows in the figure specify possible transitions between these states; all transitions between adjacent states, except the backward transition from CIN3/cancer to CIN2, were allowed. CIN3/cancer was the absorbing state. Each transition parameter λ indicates the transition intensity; i.e., λ_ij_ is interpreted as an “instantaneous risk” of transition from state i to j.

**Figure 3 cancers-12-00270-f003:**
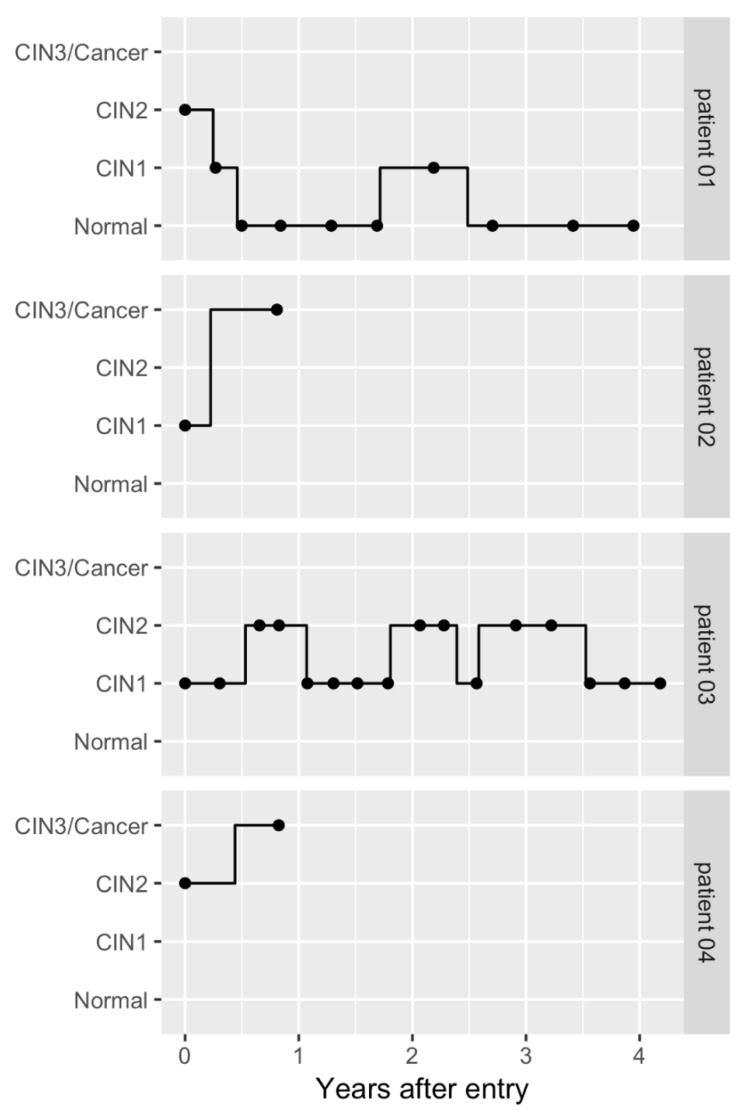
Possible transition paths for the selected patients. The figure shows possible transition paths for four selected patients (patient 01–patient 04). Cytological and histological results were combined to classify the results into the states of the model. We truncated observations after the diagnosis of cervical intraepithelial neoplasia 3 or cancer. The filled circles indicate actual observations or visits. The solid line is a possible transition path during the follow-up period. The possible transition paths were randomly selected on the basis of the observed states.

**Table 1 cancers-12-00270-t001:** Sample size and summary statistics for the combination of patients in the six HPV categories and diagnoses at the time of entry.

Diagnosis at the Time of Entry		HPV 16	HPV 18	HPV 52	HPV 58	Other hrHPVs	No hrHPVs	All
Normal	N	13	11	17	13	30	122	195
	Age at the time of entry (years), mean (SD)	42.4 (13.8)	39.5 (15.0)	36.7 (10.0)	41.3 (16.8)	42.5 (16.9)	41.2 (10.5)	41.3 (12.1)
	Number of visits, mean (SD)	5.1 (4.8)	7.2 (3.4)	7.5 (5.7)	6.8 (4.6)	7.0 (4.8)	6.5 (3.9)	6.8 (4.3)
	Follow-up interval (years), mean (SD)	0.49 (0.37)	0.46 (0.27)	0.50 (0.35)	0.49 (0.42)	0.47 (0.27)	0.53 (0.40)	0.51 (0.37)
	Follow-up period (years), mean (SD)	2.1 (2.5)	2.9 (1.6)	3.7 (3.1)	2.9 (2.3)	2.9 (2.2)	3.1 (2.1)	3.1 (2.2)
CIN1	N	23	11	38	24	79	111	259
	Age at the time of entry (years), mean (SD)	34.6 (8.2)	33.0 (10.0)	36.6 (8.6)	36.1 (7.7)	34.5 (7.2)	39.0 (10.3)	36.9 (9.2)
	Number of visits, mean (SD)	7.6 (5.3)	7.6 (4.7)	10.7 (5.9)	10.5 (6.4)	9.0 (4.9)	9.3 (5.2)	9.3 (5.2)
	Follow-up interval (years), mean (SD)	0.42 (0.43)	0.51 (0.53)	0.38 (0.18)	0.45 (0.42)	0.39 (0.22)	0.42 (0.31)	0.41 (0.29)
	Follow-up period (years), mean (SD)	3.2 (2.6)	3.2 (2.2)	4.0 (2.4)	3.9 (2.6)	3.4 (2.2)	3.9 (2.3)	3.7 (2.3)
CIN2	N	67	15	65	57	67	54	283
	Age at the time of entry (years), mean (SD)	37.6 (7.9)	42.0 (5.7)	41.2 (8.2)	39.7 (8.5)	37.9 (8.5)	36.6 (9.2)	39.1 (8.4)
	Number of visits, mean (SD)	6.8 (5.4)	7.0 (4.4)	7.2 (5.4)	8.9 (5.8)	8.6 (5.6)	8.6 (5.8)	8.0 (5.5)
	Follow-up interval (years), mean (SD)	0.35 (0.15)	0.34 (0.14)	0.38 (0.24)	0.38 (0.24)	0.39 (0.33)	0.38 (0.33)	0.37 (0.26)
	Follow-up period (years), mean (SD)	2.2 (2.2)	2.1 (1.6)	2.5 (2.2)	3.5 (2.6)	3.2 (2.3)	3.3 (2.5)	2.9 (2.3)

CIN, cervical intraepithelial neoplasia; HPV, human papillomavirus; hrHPV, high-risk human papillomavirus; SD, standard deviation. Cytological and histological results were combined to classify the results into the following diagnoses: normal, CIN1, and CIN2. HPVs 16, 18, 31, 33, 35, 39, 45, 51, 52, 56, 58, 59, and 68 were classified as hrHPVs. Of these, HPV 16, 18, 52, and 58 were categorized separately. hrHPVs other than HPVs 16, 18, 52, and 58 were classified as “other hrHPVs.” Patients who were not infected with any hrHPVs were referred to as “no hrHPVs” patients. In cases of observed coinfection with different HPV genotypes, it was possible to include the same patient in the summary statistics of multiple HPV categories.

**Table 2 cancers-12-00270-t002:** Summary of the transitions from each diagnosis of cervical epithelial lesions for the six human papillomavirus (HPV) categories.

		Diagnosis at t^th^ Visit				
Diagnosis at (t-1)^th^ Visit	HPV Category	Normal	CIN1	CIN2	CIN3	Cancer
Normal	HPV 16	206 (84.4)	13 (5.3)	21 (8.6)	4 (1.6)	0 (0.0)
	HPV 18	89 (81.6)	12 (11.0)	8 (7.3)	0 (0.0)	0 (0.0)
	HPV 52	277 (75.8)	54 (14.7)	32 (8.7)	2 (0.5)	0 (0.0)
	HPV 58	230 (80.4)	33 (11.5)	21 (7.3)	2 (0.6)	0 (0.0)
	Other hrHPVs	611 (86.1)	72 (10.1)	23 (3.2)	3 (0.4)	0 (0.0)
	No hrHPVs	1289 (90.2)	109 (7.6)	26 (1.8)	3 (0.2)	1 (0.0)
CIN1	HPV 16	29 (28.9)	34 (34.0)	35 (35.0)	2 (2.0)	0 (0.0)
	HPV 18	18 (38.2)	19 (40.4)	8 (17.0)	2 (4.2)	0 (0.0)
	HPV 52	80 (35.0)	90 (39.4)	53 (23.2)	5 (2.1)	0 (0.0)
	HPV 58	51 (31.6)	68 (42.2)	40 (24.8)	2 (1.2)	0 (0.0)
	Other hrHPVs	132 (40.7)	143 (44.1)	45 (13.8)	4 (1.2)	0 (0.0)
	No hrHPVs	203 (54.5)	132 (35.4)	34 (9.1)	3 (0.8)	0 (0.0)
CIN2	HPV 16	31 (12.1)	37 (14.4)	147 (57.4)	40 (15.6)	1 (0.3)
	HPV 18	10 (12.9)	8 (10.3)	51 (66.2)	8 (10.3)	0 (0.0)
	HPV 52	41 (13.8)	53 (17.9)	168 (56.9)	33 (11.1)	0 (0.0)
	HPV 58	32 (10.2)	45 (14.4)	210 (67.5)	24 (7.7)	0 (0.0)
	Other hrHPVs	49 (16.7)	52 (17.8)	166 (56.8)	25 (8.5)	0 (0.0)
	No hrHPVs	58 (27.2)	31 (14.5)	114 (53.5)	10 (4.6)	0 (0.0)

CIN, cervical intraepithelial neoplasia; HPV, human papillomavirus; hrHPV, high-risk human papillomavirus. Values are the number (percentage) of transitions observed from prior diagnosis to current diagnosis for each HPV category. Cytological and histological results were combined to classify the results into the following diagnoses: normal, CIN1, CIN2, CIN3, and cervical cancer. HPVs 16, 18, 31, 33, 35, 39, 45, 51, 52, 56, 58, 59, and 68 were classified as hrHPVs. Of these, HPV 16, 18, 52, and 58 were categorized separately. hrHPVs other than HPVs 16, 18, 52, and 58 were classified as “other hrHPVs.” Patients who were not infected with any hrHPVs were referred to as “no hrHPVs” patients. In cases of observed coinfection with different HPV genotypes, it was possible to include the same patient in multiple HPV categories.

**Table 3 cancers-12-00270-t003:** Predicted 2-year transition probabilities from states to states and their 95% confidence intervals (CIs) for the six HPV categories.

Current State		State after 2 Years			
	HPV Category	Normal	CIN1	CIN2	CIN3/Cancer
Normal	HPV 16	0.598 (0.506–0.684)	0.099 (0.074–0.128)	0.169 (0.127–0.215)	0.132 (0.090–0.183)
	HPV 18	0.610 (0.479–0.719)	0.156 (0.109–0.215)	0.156 (0.093–0.230)	0.076 (0.033–0.148)
	HPV 52	0.533 (0.474–0.593)	0.189 (0.162–0.219)	0.180 (0.146–0.216)	0.096 (0.070–0.130)
	HPV 58	0.559 (0.484–0.627)	0.171 (0.140–0.205)	0.206 (0.162–0.255)	0.062 (0.041–0.089)
	Other hrHPVs	0.723 (0.676–0.766)	0.155 (0.132–0.182)	0.085 (0.066–0.108)	0.034 (0.023–0.050)
	No hrHPVs	0.838 (0.814–0.861)	0.105 (0.090–0.121)	0.042 (0.032–0.054)	0.012 (0.007–0.020)
CIN1	HPV 16	0.434 (0.349–0.512)	0.089 (0.067–0.115)	0.175 (0.134–0.223)	0.300 (0.225–0.378)
	HPV 18	0.535 (0.396–0.652)	0.146 (0.100–0.207)	0.172 (0.102–0.257)	0.146 (0.069–0.267)
	HPV 52	0.473 (0.413–0.529)	0.178 (0.152–0.208)	0.183 (0.150–0.221)	0.164 (0.122–0.219)
	HPV 58	0.469 (0.399–0.535)	0.165 (0.135–0.197)	0.239 (0.192–0.291)	0.126 (0.084–0.181)
	Other hrHPVs	0.656 (0.606–0.702)	0.156 (0.133–0.181)	0.102 (0.079–0.128)	0.084 (0.058–0.119)
	No hrHPVs	0.808 (0.781–0.835)	0.107 (0.091–0.123)	0.049 (0.038–0.065)	0.034 (0.021–0.054)
CIN2	HPV 16	0.335 (0.266–0.404)	0.079 (0.059–0.101)	0.165 (0.121–0.218)	0.418 (0.330–0.512)
	HPV 18	0.373 (0.245–0.501)	0.119 (0.074–0.178)	0.186 (0.099–0.302)	0.320 (0.178–0.507)
	HPV 52	0.381 (0.324–0.434)	0.156 (0.129–0.184)	0.175 (0.138–0.216)	0.286 (0.220–0.367)
	HPV 58	0.356 (0.291–0.419)	0.150 (0.122–0.181)	0.260 (0.209–0.319)	0.232 (0.167–0.307)
	Other hrHPVs	0.518 (0.453–0.571)	0.146 (0.122–0.169)	0.117 (0.089–0.148)	0.218 (0.159–0.299)
	No hrHPVs	0.706 (0.643–0.749)	0.106 (0.090–0.123)	0.063 (0.045–0.089)	0.124 (0.079–0.191)

CIN, cervical intraepithelial neoplasia; HPV, human papillomavirus; hrHPV, high-risk human papillomavirus. Values are the predicted probabilities (95% confidence intervals) of transitions from the current state to the state after two years for each HPV category. Cytological and histological results were combined to classify the results into the following diagnoses: normal, CIN1, CIN2, CIN3, and cervical cancer. HPVs 16, 18, 31, 33, 35, 39, 45, 51, 52, 56, 58, 59, and 68 were classified as hrHPVs. Of these, HPV 16, 18, 52, and 58 were categorized separately. hrHPVs other than HPVs 16, 18, 52, and 58 were classified as “other hrHPVs.” Patients who were not infected with any hrHPVs were referred to as “no hrHPVs” patients. In cases of observed coinfection with different HPV genotypes, it was possible to include the same patient in multiple HPV categories. We used the continuous-time multistate Markov model to estimate the prognosis of each patient. We defined four states: normal (state 1), cervical intraepithelial neoplasia 1 (CIN1, state 2), CIN2 (state 3), and CIN3/cancer (state 4). Arrows in Figure 2 specify possible transitions between the states defined in our model; all transitions between adjacent states, except the backward transition from CIN3/cancer to CIN2, were allowed. CIN3/cancer was the absorbing state. We truncated observations after the diagnosis of CIN3 or cancer.

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
