# Peer review of "Multistate Markov Model to Predict the Prognosis of High-Risk Human Papillomavirus-Related Cervical Lesions"

_cancers, 2020, doi:10.3390/cancers12020270_

Round 1

Reviewer 1 Report

This paper analysed data from outpatients with abnormal cervical cytology who attended the University of Tokyo hospital in the period 2008-2015, were found to have a "normal cervical lesion", CIN1 or CIN2, and were observed for at least two visits during the study period. A multistage Markov model was applied, because it was considered that this would better account for regression as well as progression of cervical neoplasia than the Cox proportional hazards model. 

My main concerns are that was not clear to me whether patients were ascertained as a result of participation in organized cervical screening, opportunistic cervical screening, or genito-urinary symptoms that led to a Pap test being performed. Further, it was not clear whether some of the women would have been exposed to HPV vaccination. In addition, would women with HIV infection have been included? If so, might proportion have been high because of specialist nature of the University of Tokyo hospital?

I was unclear why cytological and histological results were combined (lines 257-8). Would the cytological result generate intervention that would lead to a histological result? I realize that cytological result could lead to cytological surveillance, as distinct from colposcopy and possible related interventions. In line 251, you state that patients with CIN2 were followed up without treatment, but they must have had an intervention for CIN2 to be a result - or do you mean that they had LSIL? I found the text in lines 260-265 confusing and was not sure that it matched the information in Figure 2. This lack of clarity makes the sensitivity analysis difficult to interpret.

The authors acknowledge that they did not evaluate the effect of concurrent multiple HPV infections. They state that multiple infections uncommon (line 222) but unclear whether this is based on literature or the situation in the Tokyo data.

The paper would also be strengthened if a Cox regression were done on the same data, with discussion of the similarities/differences between the multistage Markov and Cox models.

Markov assumption is that future evolution depends on the current state at time t. But isn't there a difference as to whether HPV infection is persistent, so it depends on more than state at time t? I understand that you only have information on HPV infection at time of index outpatient visit, but this could be transient or persistent. Could be argued that cytology result is a marker of persistence.

Author Response

Dear Reviewer #1,

We greatly appreciate your thoughtful and constructive comments. Please find our point-by-point responses to each of the comments (An attached file). As indicated in the responses that follow, we have addressed all your comments and suggestions during the preparation of the revised version of our manuscript.

Reviewer 2 Report

In this manuscript entitled “Multistate Markov model to predict the prognosis of high-risk human papillomavirus-related cervical lesions”, the authors applied the continuous-time multistate Markov model to the prognosis of cervical lesions based on the infected HPV types.

The authors propose that the Markov model has the potential to more accurately estimate the fates of HPV-infected cervical lesions than the standard Cox proportional hazards model because of the bidirectional nature of CIN.

According to this model, the rate of progression to CIN3 was highest in HPV 16-positive patients, followed by HPV 18-positive patients. In addition, the Markov model also revealed that HPV 52/58-positive patients tend to remain between CIN1 and CIN2 and that the other high-risk HPV types most likely regress to the normal state and least likely progress to CIN3/cancer.

The manuscript is well written and nicely presented. Overall, I think that the study provides a promising analytical method for HPV genotype-based management.

Author Response

Dear Reviewer #2,

We truly appreciate your encouraging comments. Thank you so much for reading our manuscript.

Reviewer 3 Report

MDPI Cancers

Manuscript Draft

Manuscript Number: cancers-659103

Title: Multistate Markov model to predict the prognosis of high-risk human papillomavirus-related cervical lesions

The authors applied a continuous-time Markov state-transition model to estimate the risk of developing cervical intraepithelial neoplasia (CIN) lesions by HPV-genotype category.  The authors reported that they have analyzed retrospectively the data of a cohort of patients with HPV-genotype confirmed CIN to investigate the relationship between HPV genotypes and clinical or pathological outcomes.  The main results based on this clinical cohort, were for CIN1 lesions 24–55% regression to normal, 34–44% persistence and the rest progressed to CIN2 or CIN3; for CIN2 patients, over 50% persisted and progression to CIN3 was HPV type dependent: 15.6%, 10.4%, 11.2%, 7.7%, and 8.6% for HPVs 16, 18, 52, 58, and other HPV types, respectively. Based on the predictions of the Markov model, in HPV 16-positives, 13% of the normal (no lesion), 30% of CIN1, and 42% of CIN2 progressed to CIN3+ within two years, respectively. Regression to normal histology was predicted for 43% of CIN1 and 34% of CIN2 lesions. The progression of HPV-18 infected cases were  7.1% of the normal (no lesion), 15% of CIN1, and 32% of CIN2 progressed to CIN3+ within two years, respectively.

The research topic is important and the methods used are suitable. However, there are major concerns about the validity of reported results due to incomplete reporting of the methods and assumptions that may bias results.

Major Compulsory Revisions

Methods: The methods section is placed after the discussion section p. 8. Please follow the standard guidelines for reporting and the IMRAD format.

P 8., line 242:  Since 2008, HPV genotyping has been performed in the University of Tokyo Hospital (Tokyo, Japan) for outpatients with abnormal cervical cytology… Patients were included in the study if they were (i) diagnosed with normal cervical lesion, CIN1, or CIN2 and (ii) observed for at least two visits during the study period.

This is a retrospective data analysis with potentially significant bias, as women with an abnormal cytology were followed. Classification into histologically normal (no lesion) does not mean, that those women have the same probability to develop CIN or cancer as women without a positive cytology.

4.2. Variables

p.8., line 257: Patient cytological and histological results and the date were recorded for each visit. Cytological  and histological results were combined to classify the results into any of the following four diagnoses:  normal, CIN1, CIN2, and CIN3/cancer. The investigators (experts in gynecologic oncology) convened  and determined the criteria for pathological diagnosis as follows: (1) CIN1–2 was classified into CIN2; (2) CIN2–3 and carcinoma in situ were classified into CIN3; (3) uncertain diagnoses (e.g., atypical squamous cells of uncertain significance, atypical squamous cells that cannot exclude HSIL, and dysplasia without grading) were excluded from the study due to concerns regarding diagnostic reliability; and (4) in the presence of histological and cytological examinations, the most severe classification was adopted as the final diagnosis.

When the aim of this study is to predict future CIN/cancer outcomes by hrHPV type, a simplification of the natural history of pathological development of CIN 1-3 and CIS should not be performed. The natural history should include all separate states CIN1, CIN2, CIN3/CIS and should not have an overlap (e.g. CIN2 classified in both states CIN1/2 and CIN2/CIN3+). In addition, the criteria for defining cervical lesions should be clearly stated.

p.8., line 266: The results of the HPV genotyping in cervical samples collected using swabs were recorded. Genotyping was performed once for each patient; thus, the HPV type assigned to a patient did not change over time.

As genotyping was performed only once, a change in the HPV status has not been recorded over time, which makes the prediction of future clinical status difficult (e.g. remission of hrHPV, re-infection or new infection)

4.4. Continuous-time multistate Markov model

p.8, line 285: We defined the following four states: normal (state 1), CIN1 (state 2), CIN2 (state 3), and CIN3/cancer (state 4) (Figure 2).

Authors don’t describe how they defined the different states. For example, what means “normal”? Are those women who have had a cytological abnormal result in the past (e.g. at the first visit) and no histological lesion? What is the hrHPV status of normal? In figure 2. the authors described that all transitions between adjacent states, except the backward transition from CIN3/cancer to CIN2, were allowed. CIN3/cancer was the absorbing state. However, in real life there is a probability to regress from CIN3 to CIN2 or CIN1 or even no lesion in a specific time. Therefore, this assumption is not valid and may bias the predicted progression probability from CIN2 to CIN3+. From the methods section it is not completely transparently described how the authors abstract the different transition probabilities from the observed. From the reported methods, it suggests that the authors calibrated the transition probabilities to fit observed prevalence of disease states (e.g. CIN 1) for each hrHPV type or category. Major concern: the authors do not report whether they externally validated there results based on observed data, e.g. does a full model that uses the predicted transition probabilities to simulate cervical cancer development predict observed age-specific cervical cancer in Japan? In addition, a predicted 2-year transition probability from state to state is probably too long, as regression and progression from state to state may occur in smaller time intervals. Minor comment: Although, the calibration process is clear to me. The following sentence should be revised and more clearly expressed: “Additionally, we simulated the prevalence of each state for the HPV categories in time series provided the initial prevalence of the states as same as the data.”

Results:

P4 line 126: The values for HPV-16 were already reported in the prior sentence. Therefore, this can be shortened. In addition, the numbers reported are not the same: e.g. in prior sentence the progression from CIN1 in HPV-16 was reported to be 30% and in line 126 it is 31%. Check numbers:

“The fates of HPV 18-infected cervical lesions were similar to those of HPV 16-infected cervical lesions. However, exceptions to these were the probabilities for progression to CIN3/cancer (HPV 18 vs. HPV 128 16): from the normal state, 7.1% vs. 13%; from CIN1, 15% vs. 31%; and from CIN2, 32% vs. 42%, 129 respectively.”

5 lines 150-153 describes model structure, but it should be described in detail in the methods section. Also, other sentences on this page may refer to methods used.

“We used the continuous-time multistate Markov model to estimate the prognosis of each patient. We defined 150 four states: normal (state 1), cervical intraepithelial neoplasia 1 (CIN1, state 2), CIN2 (state 3), and CIN3/cancer (state 4). Arrows in Figure 1 specify possible transitions between the states defined in our model; all transitions between adjacent states, except the backward transition from CIN3/cancer to CIN2, were allowed. CIN3/cancer was the absorbing state. We truncated observations after the diagnosis of CIN3 or cancer.”

The model predictions (Fig1.) overestimate the prevalence in the normal state and underestimate the prevalence in the CIN3 state at start and more significantly over time. This suggests that the regression to normal over time is overestimated and the progression to more severe CIN is underestimated. Internationally, Markov model transitions are calibrated to fit specific observed data including prevalence of age-specific HPV status, age-specific invasive cancer.

Tables:

Table 1: HPV16,18, 52, and 58 are also hrHPV. Therefore, for all HPV types hrHPV (or only HPV) can be used. Why do have normal women (without hrHPV) so many visits? Table 2 What means “current diagnosis” or what is the time interval between past and current diagnosis (e.g. at baseline to end of follow-up)? Please state the time horizon of the analysis -> transition from/to in which time interval? HPV16,18, 52, and 58 are also hrHPV. Therefore, for all HPV types hrHPV (or only HPV) can be used. Why is there a cancer case in women without hrHPV? And why are there CIN cases in no hrHPV? Are those probably newly-infected but not diagnosed as hrHPV-positive CIN cases? Maybe the matrix is incomplete e.g. for those regressing from normal/HPV+ to normal/No HPV Table 3 Did you compare the predicted transition probabilities with those from other Markov models simulating cervical cancer development? If so, please describe in the discussion section.

Discussion:

Minor comments

This study is not the first to apply Markov modelling for simulating future cervical cancer development. Many studies use Markov modelling (including semi Markov processes) to evaluate benefits, harms and costs of screening and prevention strategies based on modelling the natural history of cervical cancer using calibration methods to match transitions to observed data. However, mostly those observed data are from cancer registries and data for CIN progression is scarce. Cite references for consistency of results with other studies

Author Response

Dear Reviewer #3,

We greatly appreciate your thoughtful and constructive comments that have helped us remarkably. Please find our point-by-point response to each of the comments (an attached file). As indicated in the responses, we addressed all your comments and suggestions during the preparation of the revised version of our manuscript.

Round 2

Reviewer 1 Report

I appreciate responses to concerns raised on initial submission, and the addition of Cox regression analysis in particular.

Some points that still require attention in my view:

Page 8, lines 260-263: please give numbers of patients recruited by each of the two methods;

Page 8, lines 263-4 - what happens at first visit is a bit unclear - colposcopy with biopsy - do you mean loop excision if transformation zone abnormal, or punch biopsy (if so, how many? 2-4 is usual practice in many settings)? Either of these methods would provide tissue that would enable histological examination. As currently written, the text implies that colposcopy his selective on basis of histological diagnosis, which does not make sense.

Surgical intervention on patients with CIN2 - do you mean loop excision or other procedure after histological diagnosis of CIN2 on colposcopy-directed biopsy?

Page 2, lines 90-91: regarding HPV18, clarify that 2nd most common high-risk type in international data, and provide reference. 

Page 7, lines 231-2: I suggest rewriting along the following lines: "Because there are many different possible combinations of concurrent infection, the numbers of patients for each possible combination was expected to be too small for analysis  of the effect of concurrent infection"

Revise title of Table S1 to make clear that these results are from Cox regression analysis

Author Response

Dear Reviewer #1,

We greatly appreciate your thoughtful comments. Please find below our point-by-point responses to each of the comments. As indicated in the responses that follow, we have addressed all your comments and suggestions during the preparation of the revised version of our manuscript.

Comment: Page 8, lines 260-263: please give numbers of patients recruited by each of the two methods.

Response: Thank you for the comment. Although we understand that the information you pointed out will be beneficial for readers, unfortunately, we did not gather information about the source from which the patients were enrolled. In addition, this revised version needs to be submitted within 5 days and then we had no time to investigate the reasons of visit to our hospital using medical records for all patients.

Additionally, since we noticed that patients whose abnormal cytology was found in other hospitals were also enrolled, we corrected the sentence as follows:"The patients who had abnormal cytology in the population-based screening or whose abnormal cytology was found in the outpatient visits at the University of Tokyo Hospital or other hospitals were enrolled." (page 8, line 261-265)

Comment: Page 8, lines 263-4 - what happens at first visit is a bit unclear - colposcopy with biopsy - do you mean loop excision if transformation zone abnormal, or punch biopsy (if so, how many? 2-4 is usual practice in many settings)? Either of these methods would provide tissue that would enable histological examination. As currently written, the text implies that colposcopy his selective on basis of histological diagnosis, which does not make sense.

Response: We have clearly described what we performed at first visit as follows; At the first visit, we confirmed the diagnosis by performing "punch biopsy" under colposcopic examination (based on the histological diagnosis). (page 8, line 264-265)

Comment: Surgical intervention on patients with CIN2 - do you mean loop excision or other procedure after histological diagnosis of CIN2 on colposcopy-directed biopsy?

Response: Thank you for the insightful comment. Surgical intervention means conization and loop electrosurgical excision or laser vaporization. We have checked the description throughout the manuscript and coincided the description as "surgical interventions (e.g., conization and loop electrosurgical excision or laser vaporization)". (page 2, line 46-48 and page 8, line 276-277)

Comment: Page 2, lines 90-91: regarding HPV18, clarify that 2nd most common high-risk type in international data, and provide reference.

Response: Thank you for the comment. We have added a reference. (page 2, line 90)

Comment: Page 7, lines 231-2: I suggest rewriting along the following lines: "Because there are many different possible combinations of concurrent infection, the numbers of patients for each possible combination was expected to be too small for analysis of the effect of concurrent infection"

Response: Thank you for your suggestion. We have changed the sentence following your suggestion. (page 7, line 231-233)

Comment: Revise title of Table S1 to make clear that these results are from Cox regression analysis

Response: We have changed the title of Table S1 as follows; "Predicted 2-year transition probabilities from CIN1 to CIN2 or more severe lesions and their 95% confidence intervals for the six HPV categories derived from the Cox regression analysis".